# EARLY STOPPING CHAIN-OF-THOUGHTS IN LARGE LANGUAGE MODELS

## ABSTRACT

Reasoning large language models (LLMs) have demonstrated superior capacities in solving complicated problems by generating long chain-of-thoughts (CoT), but such a lengthy CoT incurs high inference costs. In this study, we introduce **ES-CoT**, an inference-time method that shortens CoT generation by detecting answer convergence and stopping early with minimal performance loss. At the end of each reasoning step, we prompt the LLM to output its current final answer, denoted as a *step answer*. We then track the run length of consecutive identical step answers as a measure of answer convergence. Once the run length exhibits a sharp increase and exceeds a minimum threshold, the generation is terminated. We provide both empirical and theoretical support for this heuristic: step answers steadily converge to the final answer, and large run-length jumps reliably mark this convergence. Experiments on five reasoning datasets across three LLMs show that ES-CoT reduces the number of inference tokens by about 41% on average while maintaining accuracy comparable to standard CoT. Further, ES-CoT integrates seamlessly with self-consistency prompting and remains robust across hyperparameter choices, highlighting it as a practical and effective approach for efficient reasoning. Implementation codes of this study are available online (hidden for peer review).

## 1 INTRODUCTION

Reasoning LLMs, such as OpenAI o-series models (OpenAI, 2024), DeepSeek R1 (Guo et al., 2025), and QwQ (Qwen-Team, 2025), have achieved state-of-the-art performance on challenging tasks in mathematics, coding, and scientific reasoning (Li et al., 2025). A key driver of this progress is chain-of-thought (CoT) reasoning, which elicits intermediate reasoning steps before producing the final answer (Wei et al., 2022). By incorporating a long thinking sequence, reasoning LLMs can plan the solution procedure, explore alternative strategies, and double-check the final result (Chen et al., 2024).

However, longer reasoning comes at a cost. For example, recent studies reveal that LLMs frequently overthink, continue to generate redundant steps even after reaching the correct answer (Chen et al., 2024). Such verbosity inflates inference cost, aggravates memory and latency challenges, and reduces the practicality of reasoning models in real-world settings. This tension motivates the study of *efficient reasoning* (Feng et al., 2025): how to preserve the accuracy benefits of CoT while minimizing unnecessary reasoning tokens.

In this work, we address this problem by asking: *When can a reasoning trajectory be stopped without harming output quality?* To answer this, we introduce the concept of a *step answer*, which is the model's current guess of the final answer at each reasoning steps. Empirical analysis across five datasets and three LLMs shows a clear convergence pattern: step answers are more likely to repeat in later reasoning stages, and at some point, this repetition length makes a sharp jump, which is a signal that the LLM is committing to a stable answer (as depicted in Figure 2). This observation aligns with prior findings that LLMs become increasingly confident as reasoning unfolds (Prystawski et al., 2023; Qian et al., 2025).

Building on this insight, we introduce **ES-CoT (Early-Stop CoT)**, an inference-time method that halts generation once a decisive convergence signal appears. Figure 1 illustrates the framework of ES-CoT. As depicted, we denote a run as a sequence of consecutive steps with the same answer,

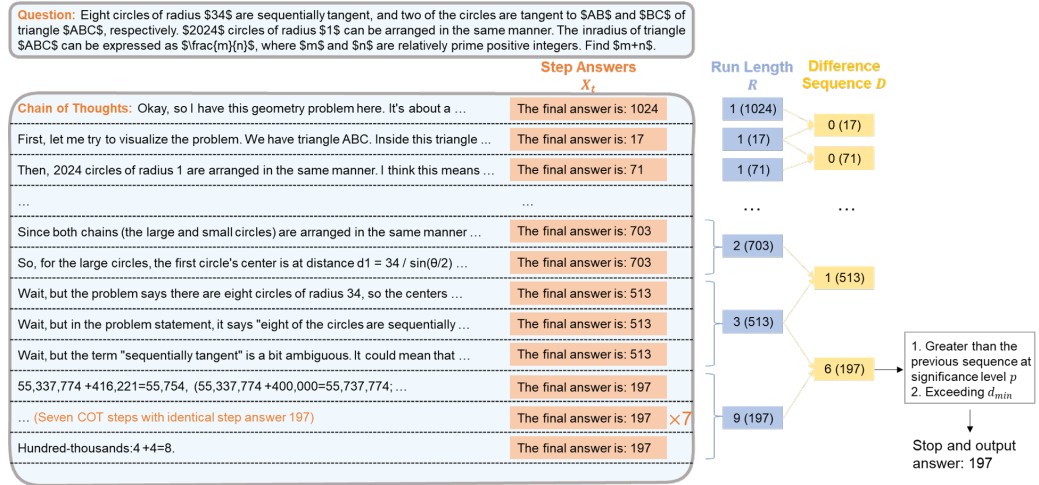

Figure 1: Framework of ES-CoT and the run-jump test

and the length of the run as the number of consecutive steps. The core of ES-CoT is the *run-jump test*: when the run length of identical step answers exhibits a statistically large leap, the reasoning is terminated, and the current step answer is returned as the final output, as shown on the right side of Figure 1. In short, ES-CoT offers a drop-in, supervision-free principle: stop thinking when the answer stabilizes. This makes efficient reasoning practical without additional models or retraining (Sui et al., 2025).

We evaluate ES-CoT across five reasoning datasets, using three LLMs of varying scales. Experimental results demonstrate that ES-CoT consistently reduces the number of generated tokens by about 41% while maintaining accuracy comparable to the original CoT prompting. Further analysis shows that ES-CoT scales robustly with different hyperparameters and integrates seamlessly with self-consistency prompting (Wang et al., 2023), yielding further gains. Our contributions are threefold:

- We propose ES-CoT, the first inference-time method that halts CoT when the run length of identical answers makes a statistically significant leap beyond previous runs. This design requires no extra reward model, no parallel decoding, and no retraining, as used in previous work (Sui et al., 2025).
- We show both empirically and theoretically that step answers converge toward the final answer, and that a sufficiently large run-length jump reliably marks this convergence.
- On five reasoning datasets with three LLMs of different scales, ES-CoT reduces token usage by about 41% on average while maintaining accuracy. Meanwhile, combined with self-consistency, ES-CoT further improves performance.

## 2 RELATED WORK

Our study belongs to the stream of efficient reasoning, which seeks to reduce reasoning length while preserving reasoning capabilities (Sui et al., 2025). Prior studies in this area can be broadly categorized into three groups: input-side, model-side, and output-side efficiency.

**Input-side (Prompt-based) efficient reasoning.** These approaches enhance reasoning efficiency by controlling the input prompt, often based on task difficulty or explicit length constraints. For instance, Chain-of-Draft (CoD) (Xu et al., 2025) encourages step-by-step reasoning but restricts verbosity by requiring each step to be expressed in no more than five words. Similarly, Token-Budget (Han et al., 2024) searches for optimal token budgets and incorporates them into prompts, thereby guiding the model to generate concise reasoning paths.

**Model-side efficient reasoning.** This stream focuses on retraining or fine-tuning models to internalize more compact reasoning strategies. O1-Pruner (Luo et al., 2025) introduces a Length-

Harmonizing Reward combined with a PPO-style optimization objective, enabling reasoning LLMs to produce shorter yet effective chains of thought (CoT). Similarly, TokenSkip (Xia et al., 2025) constructs compressed CoT data by skipping less informative tokens and fine-tunes models on these shortened trajectories, thereby encouraging more efficient internal reasoning.

**Output-side efficient reasoning.** These methods dynamically shorten reasoning during inference by adjusting the generation process. Speculative Rejection (Sun et al., 2024) leverages a reward model to estimate partial sequence quality and terminates low-quality generations early. Early Stop Self-Consistency (ESC) (Li et al., 2024) instead monitors answer convergence within a sliding window, halting generation once outputs stabilize, thus preventing unnecessary reasoning steps.

Input-side prompting methods rely on problem-specific analysis to achieve better performance, while model-side retraining requires additional training or fine-tuning of LLMs, which is often costly. In contrast, our approach belongs to output-side efficient reasoning: it adjusts reasoning length across tasks and models without extra supervision. Compared to other output-side methods (Sun et al., 2024; Li et al., 2024), ES-CoT avoids parallel decoding and eliminates the needs for auxiliary reward models. Building on empirical evidence of LLM reasoning dynamics, ES-CoT introduces the run-jump test, a simple and single-trajectory rule on answer run lengths that early stops the reasoning when the current run makes a statistically significant jump.

## 3 METHOD

### 3.1 NOTATION AND OBJECTIVE

For a target task with prompt $p_m$, let $P_M$ denote a pretrained LLM that receives the prompt and generates a solution step by step. A *step* is defined as a portion of the CoT that starts and ends with a newline character (Zheng et al., 2024). Let $T$ be the total number of steps, which is finite due to output length constraints in LLM generation.

In ES-CoT, at each step $t \in \{1, 2, ...T\}$, we append the prompt *"The final answer is"* to elicit the model's current answer (the orange column in Figure 1). We call this the *step answer* and denote its distribution as $X_t$, with $X_T$ representing the distribution of the final answer. A sample of this distribution is written as $x_t \sim X_t$. Let $\mathcal{A}$ be the answer space (the set of all possible values of $X_t$) with size $|\mathcal{A}|$.

Formally, $X_t$ should be understood as the distribution obtained by repeatedly sampling the model with the same prompt and recording the frequency of each distinct answer. This definition abstracts away from token-level likelihoods: although longer answers naturally receive lower token-level probabilities, they remain comparable to shorter answers under this frequency-based view.

The objective of early-stopping CoT is to terminate at an intermediate step $t < T$ that yields an answer similar to the final step, i.e., $X_t \approx X_T$, while keeping $t$ as small as possible to reduce inference costs.

### 3.2 CoT ANSWER DYNAMICS

We now examine how the step answer distribution evolves as $t$ increases. Experiments are conducted with three LLMs on five mathematical and logical reasoning datasets. Unless otherwise noted, results are reported as averages across all datasets. Details of the datasets and models are provided in Section 4.1.

We first conduct an empirical evaluation of the probability that the step answer matches the final answer, i.e., $P(X_t = X_T)$. We proceed as follows. (1) We first record the final answer $x_T$ for each CoT trajectory. (2) For each trajectory, we record the relative position $t/T$ whenever $x_t = x_T$. We then analyze the empirical density of these relative positions, which serves as a proxy for the probability $P(X_t = X_T)$. We depict the density distribution in Figure 2a. As shown, $P(X_t = X_T)$ increases with $t$, indicating that LLMs progressively approach the final answer as reasoning unfolds.

However, convergence alone does not provide a stopping criterion. Intuitively, $X_t$ stabilizes when multiple consecutive steps yield the same answer. To capture this, we measure the *run length* of consecutive identical answers, and denote this sequence as $R = \langle r_1, r_2, \ldots \rangle$. If $X_t$ is converging

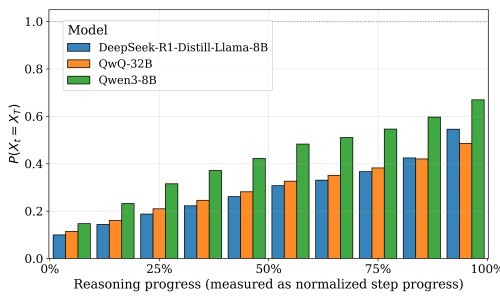 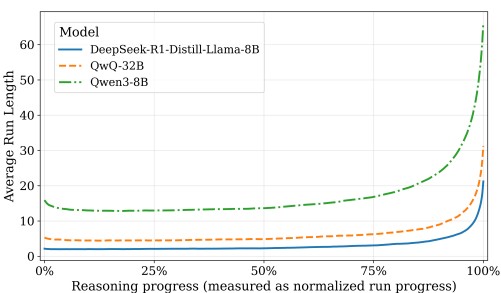

(a) Probability that step answers match the final answer, $P(X_t = X_T)$, over reasoning progress $(t/T)$. The last bar is an average over the final 10% of steps, so its value is less than one.

(b) Run lengths over reasoning progress (measured as normalized run progress). Late-stage jumps indicate converging.

Figure 2: CoT Answer Dynamics.

to $X_T$, we should observe an increasing $R$ as the model becomes more confident. For example, in Figure 1, early answers such as "1024" or "17" appear only once, while "197" repeats nine times near the end.

We plot the evolution of $R$ in Figure 2b. Unlike Figure 2a, where the x-axis tracks step progress $(t/T)$, here the x-axis tracks the index of each runs. For instance, in the run sequence $R = \langle 1, 1, 1, 2, 3, 9 \rangle$, the first three short runs occupy 3/6 of the horizontal axis, while the long final run ("197" repeated nine times) occupies the last 1/6.

The averaged results across datasets and models show that runs length grow as reasoning proceeds. More importantly, the growth curve is convex, suggesting a leap of convergence in later stages, where the model increasingly commits to a single answer. This late-stage convexity signals a decisive stopping criterion.

---

**Algorithm 1** ES-CoT

**Input:** A predefined minimum difference $d_{min}$, a pretrained LLM, and a task represented by its prompt

1: Initialize a run sequence $R = \langle r_1 \rangle$, and the run parameters $n = 1$, $r_1 = 0$
2: **for** $t = 1, 2 \ldots, T$ **do**
3:     At the end of each step $t$, add the prompt *'The final answer is'*, record the answer $x_t$
4:     **if** $x_t = x_{t-1}$ or $t = 1$ **then**
5:         $r_n \leftarrow r_n + 1$
6:     **else**
7:         $n \leftarrow n + 1$ and $r_n = 1$
8:     **end if**
9:     Update the run sequence $R$
10:    Update the difference sequence $D = \langle d_1, d_2, \ldots, d_{n-1} \rangle = \langle r_k - r_{k-1} \rangle_{k=2}^{n}$
11:    **if** $d_{n-1} > d_{min}$ and a t-test indicates that $d_{n-1}$ is significantly greater than previous differences $d_{1:n-2}$ **then**
12:        Terminate the generation, output the answer $x_t$
13:    **end if**
14: **end for**
15: Output the final answer $x_T$

**Output:** The generated answer $x_t$ or $x_T$

---

## 3.3 ES-CoT Algorithm

Building on the convergence patterns identified in Section 3.2, we formalize ES-CoT and examine its theoretical guarantees. Algorithm 1 specifies the procedure, while Section 3.4 develops conditions under which early stopping remains consistent with the final answer. Algorithm 1 maintains

a run sequence $R = \langle r_1, r_2, \ldots, r_n \rangle$ that records consecutive identical answers. To monitor how quickly runs grow, we compute the difference sequence $D = \langle d_1, d_2, \ldots, d_{n-1} \rangle$ with $d_i = r_{i+1} - r_i$. Early stopping is determined by the run-jump test: if the latest difference $d_{n-1}$ exceeds a predefined threshold $d_{min}$ and is statistically larger than the earlier differences according to a t-test, the reasoning is terminated. Otherwise, generation continues until completion. We formally define the run-jump test as follows.

**Definition 1 (Run-jump test)** *Let $R = \langle r_1, \ldots, r_n \rangle$ be the run lengths and $D = \langle d_1, \ldots, d_{n-1} \rangle$ with $d_i = r_{i+1} - r_i$. We trigger an early stop at run $n$ if*

$$d_{n-1} \geq d_{\min} \quad and \quad d_{n-1} \text{ is significantly larger than } d_{1:n-2}.$$

Intuitively, a sharp jump in run length signals a phase transition in the model's confidence: the step answer has stabilized, and further reasoning adds little value. ES-CoT halts precisely at the convergence, achieving substantial savings in inference cost without sacrificing accuracy.

### 3.4 THEORETICAL ANALYSIS OF ES-CoT

We now analyze the theoretical properties of ES-CoT. Throughout, we adopt the definitions in Section 3.1. Our focus is on regimes where the final step produces a confident answer. Tasks whose final predictions remain high-entropy are not amenable to early stopping in the first place. This is because, for tasks with high-entropy answers, even the last step may not be a stopping point. Therefore, we make the following assumption.

**Assumption 1** *(**Deterministic final answer**) The final-step answer distribution is a Dirac delta. Writing $X_T = (p_T^1, \ldots, p_T^{|\mathcal{A}|})$ over answer space $\mathcal{A}$, there exists an index $\max$ such that*

$$p_T^i = \begin{cases} 1 & if\ i = max, \\ 0 & otherwise. \end{cases} \tag{1}$$

*W.l.o.g., let $\max = 1$.*

We empirically validate Assumption 1 in Appendix A by showing that LLMs consistently produce the same final answer under different random seeds.

Next, we make an assumption about how the step-answer distributions $X_t$ evolve during reasoning. Prior work shows that the mutual information between tokens near the end of each step and the final answer token increases as reasoning progresses (Qian et al., 2025). Consistent with this, our empirical results in Figure 2a demonstrate that the probability $P(X_t = X_T)$ grows on average with $t$. Formally, we state the following assumption.

**Assumption 2** *(**Monotone approach to the final answer**) Let $p_t = P(X_t = X_T)$. We assume that $p_t$ is monotonically increasing in $t$. Under Assumption 1, this is equivalent to $p_t^1$ increasing with $t$, since $X_T$ is a point mass on answer 1 and $\Pr(X_t = X_T) = p_t^1$.*

Assumption 1 and 2 yield an explicit bound on the error of the answer obtained by ES-CoT mismatching the final answer.

**Theorem 1** *Let $k$ denote the index of the run where ES-CoT terminates early. Let $c_j = \sum_{i=1}^{j} r_i$ be the number of steps up to run $j$. For notation simplicity, denote $q = c_{k-1}$. Consider the $(k-1)$- and $k$-th runs. Let $e = P(X_{c_k} \neq X_T)$ be the error of ES-CoT, i.e., the intermediate answer obtained by ES-CoT does not match the final answer, then*

$$e \leq 1 - \frac{1}{\left( \frac{1-p_{q+1}}{p_{q+1}} \right)^{(r_k - r_{k-1})} + 1}. \tag{2}$$

*Proof. See Appendix B.1.*

**Remark 1** *Since $p_t$ is monotonically increasing and $p_T = 1$, by the squeeze theorem, there exists a half step $h \leq T$, s.t., $p_t > \frac{1}{2}$ for any $t \geq h$. If $p_t > \frac{1}{2}$, then $\frac{1-p_t}{p_t} < 1$. With a high difference in the size of runs $r_k - r_{k-1}$, the error is approaching 0.*

Theorem 1 motivates the ES-CoT design: a *large* positive jump $d_k = r_k - r_{k-1}$ is required before stopping. In Algorithm 1, this is enforced by (i) a minimum threshold $d_{min}$ (a warmup that prevents premature stops when $p_t$ is still small), and (ii) a statistical test that the latest jump is unusually large relative to previous jumps.

Theorem 1 only considers the special case where the answer space contains only 2 answers. We next extend the analysis beyond binary answer spaces by adding a mild regularity assumption on the distribution of other answers.

**Proposition 1** *Let the definition of $k, c_j, q$, and $e$ be the same as above. Suppose $|\mathcal{A}| \geq 3$ and, at each step, the distribution over* incorrect *answers is uniform.*[1] *The following inequality holds:*

$$e \leq 1 - \frac{1}{1 + (1 - p_{q+1})^{(r_k - r_{k-1})} \left(\frac{2}{|\mathcal{A}|-1}\right)^{r_k} + \left(\frac{1-p_{q+1}}{p_{q+1}} \cdot \frac{1}{|\mathcal{A}|-1}\right)^{(r_k - r_{k-1})}}. \tag{3}$$

*Proof. See Appendix B.2.*

**Remark 2** *If $q + 1$ exceeds the half step $h$ (where $p_{q+1} > \frac{1}{2}$), the upper bound in Proposition 1 goes to 0 as $r_k - r_{k-1} \to \infty$.*

Taken together, Theorem 1 and Proposition 1 formalize the intuition behind ES-CoT: once runs begin to lengthen quickly, the current answer is very likely to match the eventual final answer. Requiring a large, statistically significant jump in run length is therefore a principled stopping rule that trades a small, controllable error for substantial token savings.

## 4 EXPERIMENTS

### 4.1 EXPERIMENTAL SETUP

**Datasets.** Following Li et al. (2024), we evaluate ES-CoT on five mathematical and formal logical reasoning datasets: the American Invitational Mathematics Examination (AIME24), GPQA (Rein et al., 2024), MATH500 (Hendrycks et al., 2021), Minerva (undergraduate-level STEM) (Lewkowycz et al., 2022), and OlympiadBench (simplified as Olympiad for space concern) (He et al., 2024). For each dataset, we report two metrics: accuracy (**Acc.**) of generated answers and the average generated tokens (**Tokens #**) (Kojima et al., 2022).

**Implementation details.** We test ES-CoT on three LLMs of different scales: QwQ 32B (Qwen-Team, 2025), Qwen3 8B (Yang et al., 2025), and DeepSeek-R1-Distill-Llama 8B (Guo et al., 2025). All models are prompted using the same zero-shot templates as Kojima et al. (2022). Details on the prompts are provided in Appendix C. We fix the temperature hyperparameter at 0.6. For QwQ 32B, we additionally apply top-$p = 0.9$ and top-$k = 20$, while smaller models (Qwen3 8B and DeepSeek-R1-Distill-Llama 8B) decode without truncating. For ES-CoT, we set the minimum run-length difference to $d_{min} = 10$, and determine significance using a $t$-test with a significance p-value of 0.05.

**Evaluation protocol.** We assess ES-CoT in three stages. First, we compare it directly to standard CoT prompting (Section 4.2). Second, we integrate ES-CoT with self-consistency prompting (Wang et al., 2023) to test composability with existing decoding strategies. Third, we perform sensitivity analyses (Section 4.4) to study the robustness of ES-CoT with respect to hyperparameters.

### 4.2 MAIN RESULTS

**Token efficiency and accuracy.** We begin by comparing ES-CoT against standard CoT prompting with greedy decoding (Wei et al., 2022). Table 1 reports accuracy and the average number of generated tokens. As shown, ES-CoT reduces tokens usage by about 41% on average. For example, on the Olympiad dataset with Qwen3, greedy CoT requires 10,652 tokens per answer on average, whereas ES-CoT only needs 4,624 tokens–a reduction of 56.59%. These results demonstrate that ES-CoT delivers substantial savings in inference cost by shortening reasoning traces.

---

[1]Formally, for each $t$, conditional on $X_t \neq A_1$, we assume $\Pr(X_t = A_i \mid X_t \neq A_1) = 1/(|\mathcal{A}| - 1)$ for all $i \neq 1$.

|  |  |  | AIME | GPQA | MATH | Minerva | Olympiad |
|---|---|---|---|---|---|---|---|
| QwQ | Acc.(%)↑ | CoT | 0.63 | 0.64 | 0.60 | 0.19 | 0.35 |
|  |  | ES-CoT | 0.60 | 0.59 | 0.62 | 0.26 | 0.39 |
|  | Tokens#↓ | CoT | 12510.67 | 6946.68 | 4075.31 | 5154.74 | 9204.54 |
|  |  | ES-CoT | 8129.83 | 3623.94 | 2343.50 | 3801.81 | 4998.08 |
| Qwen3 | Acc.(%)↑ | CoT | 0.73 | 0.52 | 0.68 | 0.25 | 0.42 |
|  |  | ES-CoT | 0.50 | 0.50 | 0.62 | 0.25 | 0.35 |
|  | Tokens#↓ | CoT | 15066.57 | 7733.67 | 4750.03 | 5920.95 | 10652.43 |
|  |  | ES-CoT | 7323.40 | 4568.66 | 2332.47 | 2981.83 | 4623.99 |
| DeepSeek | Acc.(%)↑ | CoT | 0.40 | 0.43 | 0.59 | 0.16 | 0.32 |
|  |  | ES-CoT | 0.37 | 0.43 | 0.57 | 0.17 | 0.30 |
|  | Tokens#↓ | CoT | 14729.33 | 6949.40 | 3224.78 | 4136.93 | 7717.91 |
|  |  | ES-CoT | 9811.80 | 4847.46 | 2191.10 | 3761.06 | 4790.82 |

Table 1: Accuracy and average number of generated tokens across 5 datasets and 3 LLMs. Note that the number of tokens contributed by the manually added prompts in ES-CoT is also counted.

Turning to accuracy, ES-CoT achieves performance comparable to greedy CoT across all tasks. In some cases, accuracy even improves: on the Minerva and Olympiad datasets with QwQ, ES-CoT surpasses CoT despite using fewer tokens. This suggests that ES-CoT not only reduces cost but occasionally mitigates overthinking, improving answer quality (Chen et al., 2024). Overall, the results in Table 1 show that ES-CoT maintains accuracy while significantly lowering inference cost.

**Intersection of ES-CoT and CoT.** Since the objective of ES-CoT is to stop reasoning early while preserving the final answer, we further examine the overlap between ES-CoT and CoT outputs. Table 2 reports the ratio of instances where the two methods produce the same answer. Consistent with our theoretical analysis in Section 3.4, the overlap ratios are high across all models and datasets. In particular, DeepSeek yields at least 87% identical answers, confirming that ES-CoT typically halts at the point where the final answer has already stabilized.

|  | AIME | GPQA | MATH | Minerva | Olympiad |
|---|---|---|---|---|---|
| QwQ | 0.64 | 0.88 | 0.87 | 0.89 | 0.75 |
| Qwen3 | 0.78 | 0.83 | 0.85 | 0.80 | 0.71 |
| DeepSeek | 0.87 | 0.94 | 0.96 | 0.97 | 0.89 |

Table 2: The ratio of instances where ES-CoT and CoT produce the same answer.

Notably, ES-CoT is a general method that does not require any prior knowledge of model capabilities or task difficulty. The cost reduction with minimal impact on correctness stems directly from the early-stop mechanism.

### 4.3  ES-CoT with Self-consistency CoT Prompts

Self-consistency prompting (CoT+SC) (Wang et al., 2023) is a widely used decoding strategy for improving the robustness of chain-of-thought reasoning. Since ES-CoT is designed as an inference-time method, it is natural to ask whether it integrates well with such strategies. In particular, we test whether the early-stopping mechanism maintains its benefits when reasoning is performed across multiple trajectories. We therefore evaluate ES-CoT in combination with self-consistency prompting (ES-CoT+SC) (Wang et al., 2023). In this setting, the LLM generates multiple reasoning trajectories in parallel, and the final answer is determined by majority voting. We sample 10 trajectories per task.

Table 3 reports accuracy and the average number of tokens across three LLMs and five datasets. With self-consistency, ES-CoT+SC consistently outperform ES-CoT alone. For example, on DeepSeek, ES-CoT+SC improves accuracy by 11.76% on Minerva (0.17 → 0.19) and by 50% on AIME (0.40

|  |  | AIME | GPQA | MATH | Minerva | Olympiad |
|---|---|---|---|---|---|---|
| QwQ | Acc.(%)↑ ES-CoT | 0.70 | 0.65 | 0.64 | 0.24 | 0.42 |
|  | ES-CoT+SC | 0.70 | 0.65 | 0.68 | 0.29 | 0.44 |
|  | CoT+SC | 0.77 | 0.69 | 0.68 | 0.25 | 0.43 |
|  | Tokens#↓ ES-CoT | 8935.73 | 5006.70 | 2866.59 | 4576.88 | 6368.77 |
|  | ES-CoT+SC | 9460.14 | 5231.85 | 2904.99 | 4642.80 | 6172.04 |
|  | CoT+SC | 13206.05 | 7046.09 | 4074.33 | 5211.29 | 9106.79 |
| Qwen3 | Acc.(%)↑ ES-CoT | 0.70 | 0.52 | 0.65 | 0.26 | 0.39 |
|  | ES-CoT+SC | 0.63 | 0.57 | 0.67 | 0.26 | 0.41 |
|  | CoT+SC | 0.80 | 0.57 | 0.69 | 0.28 | 0.44 |
|  | Tokens#↓ ES-CoT | 9339.93 | 5605.49 | 3006.51 | 3899.26 | 6284.90 |
|  | ES-CoT+SC | 9294.25 | 5588.90 | 2972.14 | 3898.94 | 6052.52 |
|  | CoT+SC | 14637.49 | 7927.56 | 4816.98 | 6005.05 | 10499.61 |
| Deep Seek | Acc.(%)↑ ES-CoT | 0.40 | 0.43 | 0.59 | 0.17 | 0.32 |
|  | ES-CoT+SC | 0.60 | 0.51 | 0.66 | 0.19 | 0.39 |
|  | CoT+SC | 0.63 | 0.52 | 0.66 | 0.19 | 0.39 |
|  | Tokens#↓ ES-CoT | 11406.67 | 6032.63 | 2592.77 | 3938.95 | 5827.14 |
|  | ES-CoT+SC | 11079.35 | 6124.84 | 2616.94 | 4053.25 | 5946.50 |
|  | CoT+SC | 13476.21 | 7082.41 | 3321.90 | 4232.97 | 7846.47 |

Table 3: Evaluation results of ES-CoT with self-consistency prompts. The number of tokens for ES-CoT+SC and CoT+SC is reported as the average over 10 samples.

→ 0.60). Notably, the token cost per sample in ES-CoT+SC remains comparable to that of ES-CoT, demonstrating that the method scales effectively when combined with parallel decoding.

We also compare ES-CoT+SC with conventional CoT+SC. Across all datasets, ES-CoT+SC achieves similar or higher accuracy while generating fewer tokens. On DeepSeek, the reduction ranges from 4.23% (4232.97 → 4053.25 tokens) to 24.22% (7846.47 → 5946.50 tokens).

In summary, ES-CoT extends naturally to the self-consistency setting, delivering improved accuracy and substantial token savings relative to both ES-CoT and standard self-consistency CoT prompting.

## 4.4 SENSITIVITY ANALYSIS

We next study the sensitivity of ES-CoT to its two hyperparameters: the minimum run-length difference $d_{min}$ and the significance level of the t-test.

|  |  | CoT | 3 | 5 | 7 | 10 | 15 | 20 |
|---|---|---|---|---|---|---|---|---|
| QwQ | Acc.(%)↑ | 0.48 | 0.35 | 0.43 | 0.47 | 0.49 | 0.52 | 0.53 |
|  | Tokens#↓ | 7578.39 | 2351.30 | 3411.10 | 4052.59 | 4579.43 | 5171.09 | 5550.94 |
| Qwen3 | Acc.(%)↑ | 0.52 | 0.34 | 0.39 | 0.41 | 0.45 | 0.50 | 0.50 |
|  | Tokens#↓ | 8824.73 | 2870.95 | 3349.75 | 3835.72 | 4366.07 | 4952.79 | 5627.22 |
| Deep Seek | Acc.(%)↑ | 0.38 | 0.29 | 0.32 | 0.35 | 0.37 | 0.38 | 0.38 |
|  | Tokens#↓ | 7351.67 | 2777.86 | 3947.18 | 4507.65 | 5080.45 | 5671.26 | 5959.63 |

Table 4: Accuracy and average generated tokens with different $d_{min}$.

**Effect of** $d_{min}$**.** Table 4 reports accuracy and average token usage for different values of $d_{min}$ with a fixed p-value of 0.5. As $d_{min}$ grows, both accuracy and the number of generated tokens rises steadily. Intuitively, a larger threshold delays early stopping, allowing more steps before termination. In the limit, ES-CoT can match the accuracy of full CoT prompting, albeit at the cost of generating more tokens. This demonstrates that ES-CoT behaves as a scalable decoding procedure: small $d_{min}$ favors efficiency, while larger $d_{min}$ favors accuracy.

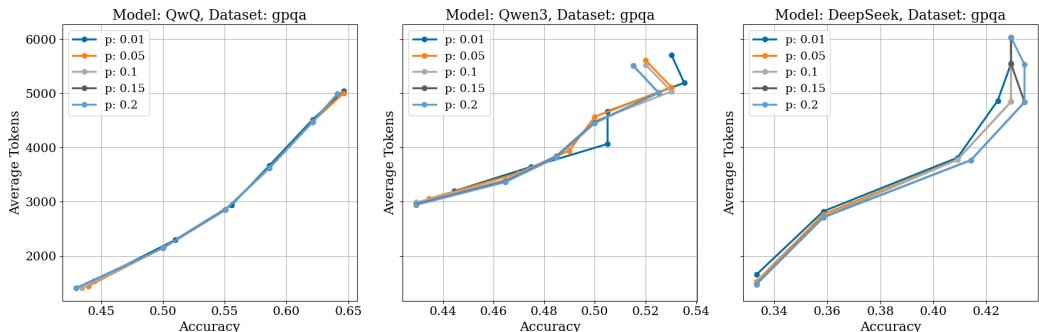

Figure 3: Robustness analysis of ES-CoT regarding the hyperparameters, including the minimum difference $d_{min}$ and p-value. For each line, we fix the p-value and vary $d_{min}$ to get the results. The results are calculated on GPQA with different LLMs.

**Effect of the t-test significance level.** We also vary the p-values threshold in Line 11 of Algorithm 1 and analyze its impact. Specifically, we vary the p-value from 0.01 to 0.2. For each fixed p-value, we set $d_{min}$ to range from 3 to 20. A lower p-value enforces stricter statistical evidence, requiring larger jumps in run length to trigger early stopping. Figure 3 shows the tradeoff between accuracy and the number of tokens on GPQA across three LLMs. As shown, for each LLM, the lines in different colors do not exhibit a significant difference. This demonstrates that ES-CoT remains robust across a wide range of p-values, indicating $d_{min}$ is the dominant parameter governing the cost-accuracy balance. We refer readers to Appendix D for results on additional datasets, where we reach similar conclusions across all datasets and LLMs.

## 5 CONCLUSIONS

In this study, we introduce ES-CoT, an inference-time method that shortens chain-of-thought reasoning while preserving answer quality. ES-CoT tracks runs of identical step answers and halts generation when the most recent run exhibits a statistically significant leap beyond prior runs and exceeds a minimum threshold. We provide empirical evidence that two patterns consistently emerge in reasoning models: (i) run length grows as reasoning proceeds, and (ii) the probability that a step answer matches the final answer increases along the trajectory. Building on these observations, we present a theoretical analysis that explains why a large jump in run length is a reliable signal for termination. Experiments on five reasoning benchmarks and three models confirmed that ES-CoT reduces token usage by about 41% on average while maintaining comparable accuracy. We also demonstrate that ES-CoT integrates well with self-consistency and remains robust across a wide range of hyperparameters.

## 6 FUTURE WORK

Looking ahead, several extensions are promising. One extension is to make ES-CoT more adaptive. For example, future work can first adjust the minimum difference and significance level based on instance-level features, then employ ES-CoT for efficient reasoning. Another direction is to broaden the evaluation to closed-source models and domains beyond mathematics and formal logic. An additional extension is to consider tasks without deterministic answers. Reducing token usage in uncertain problems remains a challenge. Finally, while ES-CoT currently focuses on convergence to the model's own final prediction, an important direction is to investigate whether early stopping can be guided by signals more directly related to the ground truth, especially in cases where the model's final prediction is incorrect.

## 7 THE USE OF LARGE LANGUAGE MODELS

LLMs are used in this study to improve the readability of the Introduction and Experiments sections.

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

## A   EMPIRICAL VALIDATION OF ASSUMPTION 1

We empirically validate Assumption 1 that the final answer distribution $X_T$ is sharply concentrated. For each problem instance, we fix the prompt and the CoT setup, then sample the model's final answer $X_T$ ten times with temperature $0.6$ and top-p $0.9$. We take the greedy-decoded final answer as the reference and compute its share among the ten samples. We then average this share over all instances within each dataset and report the averages for each model–dataset pair.

If Assumption 1 holds, the reference share should be high and close to one, indicating that the model's final answer is stable under modest stochastic sampling. As shown in Table A1, the averages are consistently high across three models and five datasets. This pattern indicates strong answer-level concentration of $X_T$ and provides direct empirical support for Assumption 1.

|  | AIME | GPQA | MATH | Minerva | Olympiad |
|---|---|---|---|---|---|
| QwQ | 1.00 | 0.98 | 0.86 | 0.85 | 0.82 |
| Qwen3 | 1.00 | 0.99 | 0.90 | 0.90 | 0.89 |
| DeepSeek | 1.00 | 0.98 | 0.87 | 0.81 | 0.83 |

Table A1: Average proportion of the greedy-decoded final answer among 10 sampled final answers (temperature $= 0.6$, top-$p = 0.9$) across datasets and models. Higher values indicate stronger stability of $X_T$, providing empirical support for Assumption 1.

## B   PROOFS

### B.1   PROOFS OF THEOREM 1

**Theorem 1** *Let $k$ denote the index of the run where ES-CoT terminates early. Let $c_j = \sum_{i=1}^{j} r_i$ be the number of steps in the first $j$ runs. For notation simplicity, denote $q = c_{k-1}$. Consider answers in the $(k-1)$- and $k$-th runs. Let $e$ be the error of ES-CoT, i.e., the intermediate answer obtained by ES-CoT does not match the final answer $e = P(X_{c_k} \neq X_T)$. If $|\mathcal{A}| = 2$, then*

$$e \leq 1 - \frac{1}{\left(\frac{1-p_{q+1}}{p_{q+1}}\right)^{(r_k - r_{k-1})} + 1}. \tag{A1}$$

**Proof.** Let $\mathcal{A}^i$ denote the $i$-th answer in the answer space, where in this case $i \in \{1, 2\}$. By definition, in the $(k-1)$-th run, one answer occurs $r_{k-1}$ times, while in the $k$-th run, a different answer occurs $r_k$ times. According to Assumption 1, $X_T = (1, 0)$ and we want $\mathcal{A}^1$ to be generated. Consider the posterior distribution after the observation, where one answer occurs $r_{k-1}$ times, then another answer occurs $r_k$ times. In the case where $|\mathcal{A}| = 2$, the only possible outcomes are $(\mathcal{A}^1, \mathcal{A}^2)$ and $(\mathcal{A}^2, \mathcal{A}^1)$. By Assumption 2, we have

$$P\left((\mathcal{A}^1, \mathcal{A}^2)\right) \leq p_{c_{k-1}}^{r_{k-1}} \cdot (1 - p_{c_{k-1}+1})^{r_k}, \tag{A2}$$

where $p_t$ is defined in Assumption 2. This is because $p_t$ increases with $t$. For generating $\mathcal{A}^1$, the maximum probability is at the last step $c_{k-1}$. For generating $\mathcal{A}^2$, the maximum probability is at the first step $c_{k-1} + 1$. For simplicity, let $q = c_{k-1}$. Inequality A2 can be re-written as

$$P\left((\mathcal{A}^1, \mathcal{A}^2)\right) \leq p_q^{r_{k-1}} \cdot (1 - p_{q+1})^{r_k}. \tag{A3}$$

Similarly, for the case of $(\mathcal{A}^2, \mathcal{A}^1)$, consider the minimum probability and we have

$$P\left((\mathcal{A}^2, \mathcal{A}^1)\right) \geq (1 - p_q)^{r_{k-1}} \cdot (p_{q+1})^{r_k}. \tag{A4}$$

With the observation, we have the conditional probability of $(\mathcal{A}^2, \mathcal{A}^1)$,

$$
\begin{aligned}
P\left(\left(\mathcal{A}^2, \mathcal{A}^1\right) |\text{Obs}\right) &= \frac{P\left(\left(\mathcal{A}^2, \mathcal{A}^1\right)\right)}{P\left(\left(\mathcal{A}^1, \mathcal{A}^2\right)\right) + P\left(\left(\mathcal{A}^2, \mathcal{A}^1\right)\right)} \\
&\geq \frac{(1-p_q)^{r_{k-1}} \cdot (p_{q+1})^{r_k}}{p_q^{r_{k-1}} \cdot (1-p_{q+1})^{r_k} + (1-p_q)^{r_{k-1}} \cdot (p_{q+1})^{r_k}} \\
&= \frac{1}{(\frac{p_q}{1-p_q})^{r_{k-1}} (\frac{1-p_{q+1}}{p_{q+1}})^{r_k} + 1} \\
&\geq \frac{1}{\left(\frac{1-p_{q+1}}{p_{q+1}}\right)^{(r_k - r_{k-1})} + 1}.
\end{aligned}
\tag{A5}
$$

The second step is because of Inequalities A3 and A4. The fourth step is because $\frac{p_q}{1-p_q} \leq \frac{p_{q+1}}{1-p_{q+1}}$.

Finally, the error is defined as

$$
e = 1 - P\left(\left(\mathcal{A}^2, \mathcal{A}^1\right) |\text{Obs}\right).
\tag{A6}
$$

This completes the proof.

### B.2 PROOFS OF PROPOSITION 1

**Proposition 1** *Let $k$ denote the index of the run where ES-CoT terminates early. Let $c_j = \sum_{i=1}^{j} r_i$ be the number of steps in the first $j$ runs and $q = c_{k-1}$. Let $e$ be the error of ES-CoT, i.e., the intermediate answer obtained by ES-CoT does not match the final answer $e = P(X_{c_k} \neq X_T)$. We further assume the answer distribution, excluding $\mathcal{A}^1$, follows a uniform distribution. Then, the following holds:*

$$
e \leq 1 - \frac{1}{1 + (1-p_{q+1})^{r_k - r_{k-1}} \left(\frac{2}{|\mathcal{A}|-1}\right)^{r_k} + \left(\frac{1-p_{q+1}}{p_{q+1}} \cdot \frac{1}{|\mathcal{A}|-1}\right)^{r_k - r_{k-1}}}.
\tag{A7}
$$

**Proof.** Similarly, we consider the posterior distribution for the observation in the $(k-1)$- and $k$-th runs. The possible outcomes are now $(\mathcal{A}^i, \mathcal{A}^1)$, $(\mathcal{A}^i, \mathcal{A}^j)$, and $(\mathcal{A}^1, \mathcal{A}^i)$, where $i \neq j$, $i \neq 1$, and $j \neq 1$. With the additional uniform assumption, we have

$$
P\left(\left(\mathcal{A}^i, \mathcal{A}^1\right)\right) \geq \left(\frac{1-p_q}{|\mathcal{A}|-1}\right)^{r_{k-1}} \cdot (p_{q+1})^{r_k},
\tag{A8}
$$

$$
P\left(\left(\mathcal{A}^i, \mathcal{A}^j\right)\right) \leq \left(\frac{1-p_{q-r_{k-1}}}{|\mathcal{A}|-1}\right)^{r_{k-1}} \left(\frac{1-p_{q+1}}{|\mathcal{A}|-1}\right)^{r_k},
\tag{A9}
$$

$$
P\left(\left(\mathcal{A}^1, \mathcal{A}^i\right)\right) \leq p_q^{r_{k-1}} \cdot \left(\frac{1-p_{q+1}}{|\mathcal{A}|-1}\right)^{r_k}.
\tag{A10}
$$

With the observation, we have the conditional probability of $(\mathcal{A}^i, \mathcal{A}^1)$,

$$
\begin{aligned}
P\left(\left(\mathcal{A}^i, \mathcal{A}^1\right) |\text{Obs}\right) &= \frac{P\left(\left(\mathcal{A}^i, \mathcal{A}^1\right)\right)}{P\left(\left(\mathcal{A}^i, \mathcal{A}^1\right)\right) + P\left(\left(\mathcal{A}^i, \mathcal{A}^j\right)\right) + P\left(\left(\mathcal{A}^1, \mathcal{A}^i\right)\right)} \\
&\geq \frac{1}{1 + \left(\frac{1-p_{q-r_{k-1}}}{1-p_q}\right)^{r_{k-1}} \left(\frac{1-p_{q+1}}{p_{q+1}}\right)^{r_k} \left(\frac{1}{|\mathcal{A}|-1}\right)^{r_k} + \left(\frac{p_q}{1-p_q}\right)^{r_{k-1}} \left(\frac{1-p_{q+1}}{p_{q+1}}\right)^{r_k} \left(\frac{1}{|\mathcal{A}|-1}\right)^{r_k - r_{k-1}}}.
\end{aligned}
\tag{A11}
$$

Let $q$ be large enough to exceed the half step $h$ (see Remark 1 for the definition of $h$). Then, for the first term in the denominator, we have

$$
\begin{aligned}
\left(\frac{1-p_{q-r_{k-1}}}{1-p_q}\right)^{r_{k-1}} & \left(\frac{1-p_{q+1}}{p_{q+1}}\right)^{r_k}\left(\frac{1}{|\mathcal{A}|-1}\right)^{r_k} \\
& \leq \left(\frac{1}{1-p_q}\right)^{r_{k-1}}\left(\frac{1-p_{q+1}}{p_{q+1}}\right)^{r_k}\left(\frac{1}{|\mathcal{A}|-1}\right)^{r_k} \\
& = \left(\frac{1-p_{q+1}}{1-p_q}\right)^{r_{k-1}}(1-p_{q+1})^{r_k-r_{k-1}}\left(\frac{1}{p_{q+1}}\right)^{r_k}\left(\frac{1}{|\mathcal{A}|-1}\right)^{r_k} \\
& \leq 1 \cdot (1-p_{q+1})^{r_k-r_{k-1}}\left(\frac{2}{|\mathcal{A}|-1}\right)^{r_k}.
\end{aligned}
\tag{A12}
$$

The first step is because $p_{q-r_{k-1}} \geq 0$. The third step is because $1-p_{q+1} \leq 1-p_q$ and $p_{q+1} \geq 0.5$. Note that, when $|\mathcal{A}| > 3$, this term is approaching 0 when $p_{q+1}$ is approaching 1 or $r_k - r_{k-1}$ is approaching $\infty$.

Next, for another term in the denominator, similar to the proof in Theorem 1, we have

$$
\left(\frac{p_q}{1-p_q}\right)^{r_{k-1}}\left(\frac{1-p_{q+1}}{p_{q+1}}\right)^{r_k}\left(\frac{1}{|\mathcal{A}|-1}\right)^{r_k-r_{k-1}} \leq \left(\frac{1-p_{q+1}}{p_{q+1}}\right)^{r_k-r_{k-1}}\left(\frac{1}{|\mathcal{A}|-1}\right)^{r_k-r_{k-1}}.
\tag{A13}
$$

When $|\mathcal{A}| > 2$, if $p_{q+1} > 0.5$, the term is approaching 0 when $r_k - r_{k-1}$ is approaching $\infty$. Combining the above two inequalities, we have

$$
P\left(\left(\mathcal{A}^i, \mathcal{A}^1\right)|\text{Obs}\right) \geq \frac{1}{1+(1-p_{q+1})^{r_k-r_{k-1}}\left(\frac{2}{|\mathcal{A}|-1}\right)^{r_k}+\left(\frac{1-p_{q+1}}{p_{q+1}}\cdot\frac{1}{|\mathcal{A}|-1}\right)^{r_k-r_{k-1}}},
\tag{A14}
$$

which completes our proof.

## C  IMPLEMENTATION DETAILS

We use the same prompt across all datasets as in (Kojima et al., 2022). Table A2 provides examples of the prompts used for different models.

| QwQ | **system**: You are a helpful and harmless assistant. You are QwQ developed by Alibaba. You should think step-by-step and put your final answer within \boxed{}. 
 **user**: A robe takes 2bolts of blue fiber and half that much white fiber. How many bolts in total does it take? |
|---|---|
| Qwen3 | **system**: You are a helpful and harmless assistant. You are Qwen developed by Alibaba. You should think step-by-step and put your final answer within \boxed{}. 
 **user**: A robe takes 2bolts of blue fiber and half that much white fiber. How many bolts in total does it take? |
| DeepSeek | **system**: You are a helpful and harmless assistant. You are DeepSeek developed by DeepSeek. You should think step-by-step and put your final answer within \boxed{}. 
 **user**: A robe takes 2bolts of blue fiber and half that much white fiber. How many bolts in total does it take? |

Table A2: Examples of prompt for different models

# D    ADDITIONAL RESULTS

In addition to the GPQA results reported in the main text (Figure 3), we conduct the same robustness analysis on other datasets and report the results in Figure A1. Overall, the results are consistency with previous findings in main text. The ES-CoT is robust to different p-values.

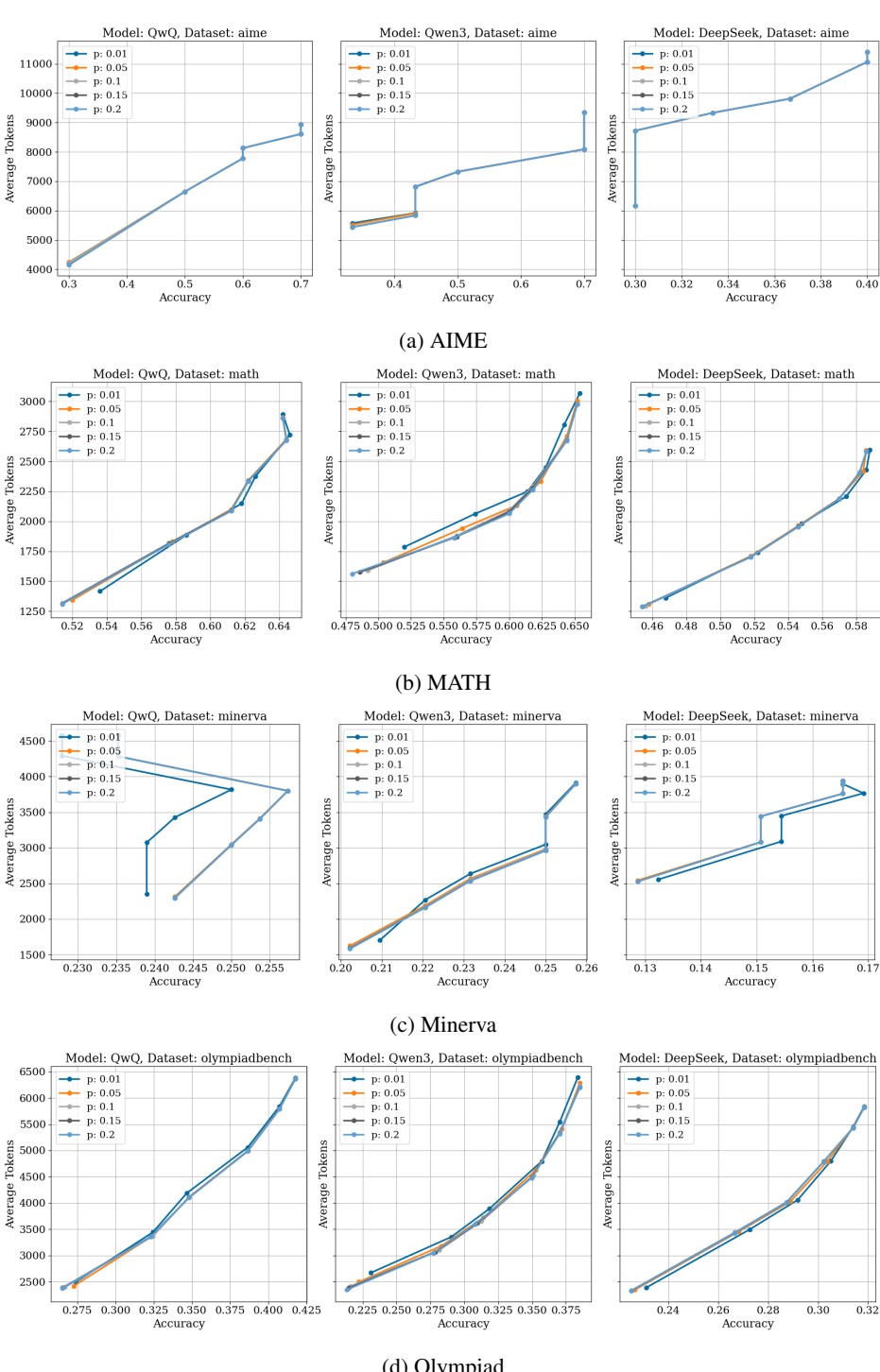

Figure A1: Robustness analysis of ES-CoT on additional datasets regarding the hyperparameters $d_{min}$ and p-value.

