# OpenReview forum: "Early Stopping Chain-of-thoughts in Large Language Models"
_ICLR.cc/2026/Conference — Submitted to ICLR 2026_

### Official Review · Reviewer_kr89 · 2025-10-27

**Soundness:** 3
**Presentation:** 3
**Contribution:** 2
**Rating:** 4
**Confidence:** 2

**Summary:**

The paper proposes ES-CoT, an inference-time method that reduces CoT length by detecting answer convergence and stopping generation early. By monitoring repeated step answers, the method terminates once convergence stabilizes, cutting inference tokens. Experiments on multiple datasets and LLMs show ES-secondo me CoT’s efficiency, robustness, and compatibility with self-consistency prompting.

**Strengths:**

- Overall, the paper is well-written.
- Efficiency in LLMs is an important and valid research challenge.

**Weaknesses:**

- Several related works on concise reasoning and CoT compression are missing from the discussion and experiments. The authors should compare or at least discuss their approach. Some of them (for example):
- Xu et al. – Chain of Draft: Thinking Faster by Writing Less
- Aytes et al. – Sketch-of-Thought: Efficient LLM Reasoning with Adaptive Cognitive-Inspired Sketching
- Zhang et al. – LightThinker: Thinking Step-by-Step Compression
- Fatemi et al. – Concise Reasoning via Reinforcement Learning
- Lee et al. – How Well Do LLMs Compress Their Own Chain-of-Thought? A Token Complexity Approach
- Nayab et al. - Concise Thoughts: Impact of Output Length on LLM Reasoning and Cost

Although the proposed method reduces long CoT generation, it still requires multiple-step answers. The authors should clarify how these results lead to a significant reduction in inference cost.

The mathematical framework appears somewhat isolated. The authors should better explain their technical contribution and how it supports or strengthens the experimental findings.

**Questions:**

Please address computational cost and theoretical issues indicated in the weaknesses section.

---

> ### Author Response · Authors · 2025-11-30
>
> The authors thank the reviewer for providing us with valuable suggestions.
>
> ```
> [Q1] Benchmarks.
> ```
>
> Thank you for your valuable comment. Since our ES-CoT method is an inference-stage method that adds a prompt manually and records step answers, ES-CoT does not conflict with the prompt engineering methods [1,2] and fine-tuning-based methods [3] in efficient reasoning. Therefore, we can combine prompt engineering and ES-CoT as in the evaluation of the combination of self-consistency prompts and ES-CoT in section 4.3. To this end, we incorporate ES-CoT with chain-of-draft (CoD) and concise CoT (CCoT) prompts, respectively, and evaluate their performances. The hyperparameters are set the same as in the manuscript. We leave the combination of ES-CoT and fine-tuning-based efficient reasoning models for future work. The results are shown in the following table with the AIME dataset generated by QwQ-32B.
>
> |Method|Accuracy|Number of tokens|
> |-|:-:|:-:|
> | CoT | 0.63 | 12510.67  |
> | ES-CoT | 0.60 | 8129.83 |
> | CoD | 0.58 | 11372.63  |
> | ES-CoT+CoD | 0.53 | 6709.40  |
> | CCoT | 0.53 | 10824.97  |
> | ES-CoT+CCoT | 0.53 | 7575.13  |
>
> Overall, the results in the table consistently show that ES-CoT maintains accuracy while significantly lowering inference cost with different prompt engineering baselines.
>
> ```
> [Q2] Although the proposed method reduces long CoT generation, it still requires multiple-step answers. The authors should clarify how these results lead to a significant reduction in inference cost.
>
> The mathematical framework appears somewhat isolated. The authors should better explain their technical contribution and how it supports or strengthens the experimental findings.
> ```
>
> Thank you for your comment. The underlying rationale of ES-CoT is to dynamically monitor the reasoning trace, typically by the extracted step answers. Since standard CoT trace usually incurs a long generation time, our method can identify the stopping point when the model is confident enough to output the final answer.
>
> The mathematical framework describes the intuition behind ES-CoT. If $r_k-r_{k-1}$ is large enough (the run length increases significantly), the current answer is very likely to match the eventual final answer. This inspires the ES-CoT method, which shows that the early stopped answer by this run jump test has a high probability of matching the final answer. We will elaborate more on the intuition of the theoretical framework in our revised manuscript.
>
> [1] Xu, Silei, et al. "Chain of draft: Thinking faster by writing less." arXiv preprint arXiv:2502.18600 (2025).
>
> [2] Nayab, Sania, et al. "Concise thoughts: Impact of output length on llm reasoning and cost." arXiv preprint arXiv:2407.19825 (2024).
>
> [3] Fatemi, Mehdi, et al. "Concise reasoning via reinforcement learning." arXiv preprint arXiv:2504.05185 (2025).

---

### Official Review · Reviewer_oxFa · 2025-10-29

**Soundness:** 3
**Presentation:** 3
**Contribution:** 3
**Rating:** 6
**Confidence:** 2

**Summary:**

This paper introduces ES-CoT, an inference-time method to reduce chain-of-thought (CoT) reasoning length in large language models (LLMs) by stopping generation early when answer convergence is detected. ES-CoT monitors the consecutive run length of identical 'step answers' (the model's current statements of its answer at each reasoning step) and halts generation using a run-jump test—triggered when the run length makes a statistically significant increase, exceeding a minimum threshold. The paper justifies this heuristic empirically and theoretically, demonstrating that step answers typically stabilize before completion. Experiments on five reasoning datasets and three LLMs show ES-CoT reduces token usage by about 41% on average with minimal loss in accuracy, and is robust across hyperparameter settings and compatible.

**Strengths:**

(1)The paper provides a quantitative and visual analysis of answer convergence in LLM reasoning (see Figure 2 and Figure 3), demonstrating that step answers indeed tend to stabilize toward the end of reasoning trajectories. The convex jump in run length (Figure 3) offers a clear, observable signal for potential early stopping.
(2)The ES-CoT mechanism is straightforward to implement at inference, requiring only minor modifications (adding a 'final answer' prompt per step), and does not require retraining, parallel decoding, or auxiliary models.
(3)The method is evaluated across three competitive LLMs and five well-established reasoning benchmarks, showing substantial savings in average tokens (see Table 1) while maintaining comparable or sometimes even better accuracy.

**Weaknesses:**

(1):The paper does not provide head-to-head results with Speculative Rejection (Sun et al., 2024) or Early Stop Self-Consistency (ESC, Li et al., 2024), which also propose output-side early stopping methods for CoT. These missing baselines are critical for evaluating the genuine advantage of ES-CoT. Without this, it is unclear if the proposed method achieves better cost-quality tradeoffs or simply offers a simpler alternative.
(2) There is no systematic examination of what types of questions or failure modes lead to suboptimal early stopping (incorrect halts, divergence from the true answer, etc.). A deeper dive into breakdowns by problem type or error case is warranted for practical adoption.
(3) ES-CoT assumes the 'step answer' extraction at each reasoning step can be done reliably. In real-world settings, especially with less structured or conversational prompts, it may be challenging to parse step-wise answers automatically, limiting applicability to domains outside structured problem-solving.

**Questions:**

(1):How would ES-CoT perform or adapt in tasks where answer uncertainty remains high, or where step answers do not stabilize (violation of Assumption 1)? Can the authors provide breakdowns or error analysis for such cases?
(2):Is the t-test actually an appropriate statistical test on the run difference sequence $D$? Have the authors considered non-parametric or more tailored sequential change-detection approaches?
(3):How robust is the step-answer extraction in less-structured CoT outputs, e.g., open-domain QA or conversation? Can ES-CoT be reasonably applied in those settings?
(4):For the token counting, are the per-step answer prompts included in both ES-CoT and baseline methods? How does ES-CoT’s overhead affect the true token savings?

---

> ### Author Response · Authors · 2025-11-30
>
> The authors thank the reviewer very much for the comments.
>
> ```
> [Q1] How would ES-CoT perform or adapt in tasks where answer uncertainty remains high, or where step answers do not stabilize (violation of Assumption 1)? Can the authors provide breakdowns or error analysis for such cases?
> ```
>
> Thank you for your comments. We can relax Assumption 1 by only requiring that the final solution has a higher probability than any other answer. Under this weaker condition, similar bounds can still be derived, although the probability of early-stop errors increases as the margin between the probability of the final solution and that of other solutions narrows. We will include a detailed discussion of this relaxed assumption and its implications in the appendix.
>
> ```
> [Q2] Is the t-test actually an appropriate statistical test on the run difference sequence $D$? Have the authors considered non-parametric or more tailored sequential change-detection approaches?
> ```
>
> We appreciate your valuable comment. The t-test we utilize is actually the sequential t-test [1], which is naturally a sequential change-detection test and tests if a sequence changes significantly on average. In our study, the sequence is the difference of run lengths.
>
> ```
> [Q3] How robust is the step-answer extraction in less-structured CoT outputs, e.g., open-domain QA or conversation? Can ES-CoT be reasonably applied in those settings?
> ```
>
> Following your comment, we also test our ES-CoT method on the CommonsenseQA dataset [1]. We set $d_{min}=10$ and $p\text{-value}=0.5$ as in our manuscript. The results are shown in the following table. These results also demonstrate that ES-CoT delivers savings in inference cost while maintaining accuracy.
>
> |Method|Accuracy|Number of tokens|
> |-|:-:|:-:|
> | CoT | 0.85 | 639.65 |
> | ES-CoT | 0.85 | 565.72 |
>
> ```
> [Q4] For the token counting, are the per-step answer prompts included in both ES-CoT and baseline methods? How does ES-CoT's overhead affect the true token savings?
> ```
>
> Thank you for your comment. The per-step prompts are not included in the token counting in tables. Although ES-CoT introduces a small overhead (the inserted prompt is about 5 tokens per step), the total number of steps is typically between 50 and 100, so the maximum additional cost is under 500 tokens. Even after accounting for this overhead, ES-CoT still yields a substantial token reduction.
>
> [1] Wald, A. (2004). Sequential analysis. Courier Corporation.
>
> [2] Talmor, Alon, et al. "Commonsenseqa: A question answering challenge targeting commonsense knowledge." Proceedings of the 2019 Conference of the North American Chapter of the Association for Computational Linguistics: Human Language Technologies, Volume 1 (Long and Short Papers). 2019.

---

### Official Review · Reviewer_r9Q5 · 2025-10-31

**Soundness:** 3
**Presentation:** 3
**Contribution:** 2
**Rating:** 2
**Confidence:** 4

**Summary:**

The paper proposes early-stopping CoT (ES-CoT), an inference-time method that aims to reduce reasoning length by producing a final answer at every reasoning step. The authors introduce the concept of run length, which is the number of consecutive steps yielding the same final answer, and use it to detect answer convergence. Both empirical and theoretical analyses are provided to justify run length as a reliable convergence signal. Experiments on five mathematical reasoning datasets show that ES-CoT significantly reduces the number of inference tokens.

**Strengths:**

- The paper is easy to read.
- Based on empirical findings, authors make two assumptions: 1) final answers are deterministic, 2) the probability of an intermediate answer being the same as the final answer monotonically increases as the reasoning progresses. Based on the two assumptions, the authors provide a theoretical justification of the run length based early stopping CoT method, which is sound.
- Authors provide sensitivity and robustness analysis of their method.

**Weaknesses:**

- Experiments focus on mathematical (or formal logic) reasoning datasets.
The generalizability of the method is unclear since the core assumptions (e.g., deterministic final answers) may not hold in other domains such as commonsense or open-ended reasoning.
- Accuracy is not preserved. In Table 1, ES-CoT outperforms the CoT baseline in only 4 out of 20 cases, with 3 of those improvements observed on the larger QwQ 32B model. The method performs poorly on smaller models. While token usage is reduced, this often comes at a substantial cost to accuracy (e.g., on AIME with Qwen3, accuracy drops from 0.73 to 0.50).
- Even when combined with self-consistency prompting, ES-CoT underperforms the baseline (CoT + SC) in nearly all cases.
- The method introduces an additional hyperparameter—the difference threshold between run lengths—yet the paper provides no clear guidance on how to select this parameter. The performance seems to vary a lot depending on this parameter.

**Questions:**

In Figure 2-(a), why is the probability that step answers match the final answer so low across all models, even at 100% reasoning progress?

---

> ### Author Response · Authors · 2025-11-30
>
> The authors thank the reviewer for providing us with valuable suggestions.
>
> ```
> [Q1] Experiments focus on mathematical (or formal logic) reasoning datasets. The generalizability of the method is unclear since the core assumptions (e.g., deterministic final answers) may not hold in other domains such as commonsense or open-ended reasoning.
> ```
>
> We acknowledge the limitations of this study, particularly the requirement of a deterministic answer for step-answer extraction. However, open-ended tasks are inherently difficult to evaluate, and developing effective evaluation for them remains an problem in the broader efficient-reasoning literature.
>
> We also test our ES-CoT method on the CommonsenseQA dataset [1], which is an open-ended dataset but has deterministic answers. We set $d_{min}=10$ and $p\text{-value}=0.05$ consistent with the setting in our manuscript. The results are shown in the following table. These findings further demonstrate that ES-CoT reduces inference cost while maintaining accuracy.
>
> |Method|Accuracy|Number of tokens|
> |-|:-:|:-:|
> | CoT | 0.85 | 639.65 |
> | ES-CoT | 0.85 | 565.72 |
>
> ```
> [Q2] Accuracy is not preserved. In Table 1, ES-CoT outperforms the CoT baseline in only 4 out of 20 cases, with 3 of those improvements observed on the larger QwQ 32B model. The method performs poorly on smaller models. While token usage is reduced, this often comes at a substantial cost to accuracy (e.g., on AIME with Qwen3, accuracy drops from 0.73 to 0.50).
> ```
>
> Thank you for your comments. For a fixed reasoning LLM, a tradeoff between token length and accuracy is inherent. That is, a certain accuracy drop is unavoidable when substantially reducing reasoning tokens [2,3,4]. Our goal is to achieve large token savings with minimal loss in accuracy, rather than surpassing full-length CoT in terms of all metrics.
>
> We also observe the performance gap between small and large models which we attribute to the stronger reasoning capabilities of large models. In our experiment, we find DeepSeek (8B) generally underperforms other two models. This holds for all methods including CoT and ES-CoT. Therefore, choosing the base model influences the overall result and can explain the relatively poor performance of our method on smaller mdoels.
>
> On the other hand, we believe that large models are more prone to overthinking. Our method can mitigate this issue, thereby improving performance. The accuracy improvement achieved by ES-CoT primarily occur in large models, which supports our claim.
>
> ```
> [Q3] Even when combined with self-consistency prompting, ES-CoT underperforms the baseline (CoT + SC) in nearly all cases.
> ```
>
> We apologize for the confusion regarding Table 3. The purpose of this table is to illustrate the generalizability of ES-CoT. Therefore, we apply self-consistency (SC) on top of ES-CoT. The key comparison in Table 3 is between ES-CoT+SC and CoT+SC, where ES-CoT+SC achieves substantial token savings with only a minor accuracy reduction. We also include ES-CoT in this table to show the accuracy improvement obtained when moving from ES-CoT to ES-CoT+SC. Taken together, these results demonstrate that ES-CoT is easy to integrate with other reasoning prompting strategies (such as SC prompts).

---

> > ### Author Response · Authors · 2025-11-30
> >
> > ```
> > [Q4] The method introduces an additional hyperparameter--the difference threshold between run lengths--yet the paper provides no clear guidance on how to select this parameter. The performance seems to vary a lot depending on this parameter.
> > ```
> >
> > Thank you for your comments. In Section 4.4, we conduct the sensitivity analysis on the parameters. From Figure 3, we find that the significance level of the run-jump t-test is not sensitive, because lines with all colors yield similar performances in each subfigure. However, when selecting $d_{min}$, small $d_{min}$ favors efficiency, while larger $d_{min}$ favors accuracy. A clear tradeoff between the number of reduced tokens and accuracy is observed. We acknowledge that selecting the proper $d_{min}$ requires additional adjustment in practice.
> >
> > ```
> > [Q5] In Figure 2-(a), why is the probability that step answers match the final answer so low across all models, even at 100% reasoning progress?
> > ```
> >
> > The probability does not reach 1 at the end because in Figure 2(a), the final bar corresponds to the 90%-100% progress bin rather than exactly 100%, so it averages over these intermediate steps (90%-100%), where the answer may not match the final solution. As a result, the value of the last bar is below one.
> >
> > [1] Talmor, Alon, et al. "Commonsenseqa: A question answering challenge targeting commonsense knowledge." Proceedings of the 2019 Conference of the North American Chapter of the Association for Computational Linguistics: Human Language Technologies, Volume 1 (Long and Short Papers). 2019.
> >
> > [2] Lee, Ayeong, Ethan Che, and Tianyi Peng. "How well do llms compress their own chain-of-thought? a token complexity approach." arXiv preprint arXiv:2503.01141 (2025).
> >
> > [3] Xu, Silei, et al. "Chain of draft: Thinking faster by writing less." arXiv preprint arXiv:2502.18600 (2025).
> >
> > [4] Nayab, Sania, et al. "Concise thoughts: Impact of output length on llm reasoning and cost." arXiv preprint arXiv:2407.19825 (2024).

---

### Official Review · Reviewer_JTng · 2025-10-31

**Soundness:** 1
**Presentation:** 2
**Contribution:** 2
**Rating:** 4
**Confidence:** 4

**Summary:**

This paper presents ES-CoT (Early-Stop CoT), an inference-time method that lets a LLM quit early while it is still doing CoT reasoning. At every step the authors force the model to output its current final answer (step answer). If that answer repeats for a suddenly longer run than before, the process stops and returns that answer. No extra training, reward model, or parallel runs are needed. Tests on 5 math/logic datasets (AIME, GPQA, MATH, Minerva and Olympiad) and 3 LLMs (QwQ 32B, Qwen3 8B, DeepSeek-R1-Distill-Llama 8B) show 41 % fewer generated tokens on average while the accuracy is close to that of standard CoT. Theory and ablations are supplied and the code is promised.

**Strengths:**

1. ES-CoT needs no re-training or extra GPUs, so it's convenient to reproduce the results with LLMs.
2. The stopping signal (the "jump" in the run-length of identical step-answers) is statistically testable, giving experiments a clear, theory-backed criterion.
3. Across 5 mathematical/logical datasets and 5 LLMs the method reduces 41% of generated tokens on average, and in several cases even raises accuracy by preventing over-thinking.
4. When paired with self-consistency, decoding ES-CoT keeps most of its token saving and sometimes yields extra accuracy, showing that the method of Early-Stop is useful to some extent.
5. Extensive ablation on thresholds and p-value shows stable behaviour, and the failure risk is low and controllable.

**Weaknesses:**

1. Every reasoning step is interrupted with the manually inserted prompt "The final answer is", which increases prompt tokens and may make some models toward stopping reasoning too early.
2. The datasets used in this paper have deterministic final answers and open-ended, creative tasks are excluded, which may lead to insufficient validation of ES-CoT's universal applicability.
3. Since no comparison is made against distilled, same-scale short-CoT LLMs that were explicitly trained for shorter reasoning, it remains unclear how much of the 41% token savings is simply due to weak baselines.
4. ES-CoT does not achieve higher reasoning accuracy than standard CoT in most experiments, so despite its efficiency in token usage, there is limited evidence for its substantive effectiveness in reasoning.
5. The high temperature (0.6) may introduce excessive randomness, and the limited model scales tested (only 32B and 8B) restrict the generalizability of the findings. These factors weaken the reliability and applicability of the conclusions.
6. The tables are poorly formatted. For instance, in Tables 3 and 4, the word "DeepSeek" is split across rows, and there are no annotations to draw attention to significant data differences.

**Questions:**

See the weakness above.

---

> ### Author Response · Authors · 2025-11-30
>
> We thank the reviewer very much for providing us with valuable suggestions to discuss the improvement of our method.
>
> ```
> [Q1] Every reasoning step is interrupted with the manually inserted prompt "The final answer is", which increases prompt tokens and may make some models toward stopping reasoning too early.
> ```
>
> We are sorry for the confusion. After we insert the prompt "The final answer is" and get the step answer during step-by-step reasoning, the model continues generating from the text before that inserted prompt. Therefore, adding the prompt helps us get the step answer, but does not directly cause early stopping. We will update Figure 1 and Algorithm 1 to be more precise.
>
> Meanwhile, the manually added prompt does not affect the generation efficiency. This is because such prompts are fixed. Therefore, inserting this prompt incurs only $O(1)$ additional generation complexity and adds just four tokens to the KV cache. To this end, we only count the tokens generated by the model itself, excluding these manually inserted prompts.
>
> ```
> [Q2] The datasets used in this paper have deterministic final answers and open-ended, creative tasks are excluded, which may lead to insufficient validation of ES-CoT's universal applicability.
> ```
>
> We acknowledge the limitations of this study, particularly the requirement of a deterministic answer for step-answer extraction. However, open-ended tasks are inherently difficult to evaluate, and developing effective evaluation for them remains an problem in the broader efficient-reasoning literature.
>
> Following your comment, we also test our ES-CoT method on the CommonsenseQA dataset [1], which is an open-ended dataset but has deterministic answers. We set $d_{min}=10$ and $p\text{-value}=0.05$ consistent with the setting in our manuscript. The results are shown in the following table. These findings further demonstrate that ES-CoT reduces inference cost while maintaining accuracy.
>
> |Method|Accuracy|Number of tokens|
> |-|:-:|:-:|
> | CoT | 0.85 | 639.65 |
> | ES-CoT | 0.85 | 565.72 |
>
> ```
> [Q3] Since no comparison is made against distilled, same-scale short-CoT LLMs that were explicitly trained for shorter reasoning, it remains unclear how much of the 41% token savings is simply due to weak baselines.
> ```
>
> Thank you for your comment. Note that our method requires no training, which makes it far more cost-efficient than approaches that rely on distilled models. Moreover, such specially short-CoT models often suffer from alignment tax, where reducing the number of tokens comes at the expense of diminished reasoning capability [2].
>
> Meanwhile, our approach is an inference-stage method and is therefore compatible with distilled or short-CoT LLMs. When applied to different models, we can still achieve token reductions without compromising accuracy.

---

> > ### Author Response · Authors · 2025-11-30
> >
> > ```
> > [Q4] ES-CoT does not achieve higher reasoning accuracy than standard CoT in most experiments, so despite its efficiency in token usage, there is limited evidence for its substantive effectiveness in reasoning.
> > ```
> >
> > For a fixed reasoning LLM, a tradeoff between token length and accuracy is inherent. That is, a certain accuracy drop is unavoidable when reducing reasoning tokens, which is a common tradeoff found in the literature [3,4,5]. Our goal is to achieve large token savings with minimal loss in accuracy, rather than surpassing full-length CoT in terms of all metrics.
> >
> > However, as shown in Table 1 in Section 4.2, ES-CoT achieves comparable performance to CoT across nearly all tasks. In some cases, ES-CoT even outperforms standard CoT. This shows that ES-CoT sometimes mitigates overthinking, which is not achieved by previous work.
> >
> > ```
> > [Q5] The high temperature (0.6) may introduce excessive randomness, and the limited model scales tested (only 32B and 8B) restrict the generalizability of the findings. These factors weaken the reliability and applicability of the conclusions.
> > ```
> >
> > Thank you for your comments. We set the temperature to 0.6 because this value is recommended on the official QwQ website. We also evaluate the performance under different randomness levels (temperature = 1 and 0.3) on the AIME dataset generated by QwQ-32B. As shown in the table below, the results are consistent across different temperatures. For standard reasoning LLMs, we believe models in the 8B-32B range are widely used. We leave the evaluation of larger models for future work.
> >
> > |Method|Accuracy|Number of tokens|
> > |-|:-:|:-:|
> > | CoT($\tau=1$) | 0.63 | 13910.30 |
> > | ES-CoT($\tau=1$) | 0.57 | 8825.07 |
> > | CoT($\tau=0.6$) | 0.63 | 12510.67  |
> > | ES-CoT($\tau=0.6$) | 0.60 | 8129.83 |
> > | CoT($\tau=0.3$) | 0.63 | 12743.73  |
> > | ES-CoT($\tau=0.3$) | 0.63 | 7458.20 |
> >
> > ```
> > [Q6] The tables are poorly formatted. For instance, in Tables 3 and 4, the word "DeepSeek" is split across rows, and there are no annotations to draw attention to significant data differences.
> > ```
> >
> > Thank you for your comments. We will improve the format of the tables in our future manuscripts.
> >
> > [1] Talmor, Alon, et al. "Commonsenseqa: A question answering challenge targeting commonsense knowledge." Proceedings of the 2019 Conference of the North American Chapter of the Association for Computational Linguistics: Human Language Technologies, Volume 1 (Long and Short Papers). 2019.
> >
> > [2] Lin, Yong, et al. "Mitigating the alignment tax of rlhf." Proceedings of the 2024 Conference on Empirical Methods in Natural Language Processing. 2024.
> >
> > [3] Lee, Ayeong, Ethan Che, and Tianyi Peng. "How well do llms compress their own chain-of-thought? a token complexity approach." arXiv preprint arXiv:2503.01141 (2025).
> >
> > [4] Xu, Silei, et al. "Chain of draft: Thinking faster by writing less." arXiv preprint arXiv:2502.18600 (2025).
> >
> > [5] Nayab, Sania, et al. "Concise thoughts: Impact of output length on llm reasoning and cost." arXiv preprint arXiv:2407.19825 (2024).

---

### Meta-Review · Area_Chair_AknF · 2026-01-06

**Summary:**

This paper introduces ES-CoT, an inference-time technique designed to reduce the computational cost of chain-of-thought reasoning in large language models by detecting answer convergence and halting generation early. Reviewers acknowledged that the method yields non-trivial token savings on mathematical reasoning tasks and is straightforward to implement. However, they raised significant concerns regarding its limited generality beyond deterministic-answer domains, the frequent accuracy degradation especially on smaller models, the absence of comparisons with closely related early-stopping approaches, and the insufficient integration between the theoretical framework and empirical findings. While the authors provided thoughtful responses, the core limitations concerning applicability and competitive evaluation remain unresolved. Given these issues, I recommend rejecting the submission.

**Reviewer Concerns:**

Based on the author rebuttal, several specific reviewer concerns were directly addressed. The authors clarified the minimal overhead of the inserted prompt, provided additional experiments on the CommonsenseQA dataset to partially address generalizability, and demonstrated compatibility with other prompt-based methods like Chain-of-Draft. They also supplied results across different temperature settings and offered clarification on the statistical test used.

However, significant concerns remain outstanding. The core limitation regarding the method's applicability to open-ended or creative tasks persists, as the additional test still uses a dataset with a deterministic answer. The frequently observed degradation in accuracy, especially for smaller models, was not resolved but rather framed as an inherent trade-off. Crucially, the lack of comparison against key contemporary baselines like Speculative Rejection or Early Stop Self-Consistency leaves the paper's contribution and relative advantage unclear. Furthermore, concerns about the practical selection of the run-length threshold and the integration between the theoretical framework and empirical findings were not adequately mitigated. These unresolved issues pertain directly to the novelty, rigor, and practical utility of the proposed method.

**Reviewer Scores:**

| Reviewer | Initial Score | Predicted New Score | Rationale for Score Change |
| - | - | - | - |
| JTng | 4 | 4 | The rebuttal clarified some implementation details and tested an additional dataset. However, core concerns about limited generalizability to open-ended tasks and the lack of comparison to key baselines (e.g., distilled/short-CoT models) were not substantively addressed. The methodological scope and central weaknesses remain. |
| r9Q5 | 2 | 2 | The authors' responses did not resolve the primary criticism of frequent accuracy degradation, especially on smaller models. The rebuttal framed this as an inherent trade-off, which is unlikely to satisfy a reviewer who views it as a major flaw. The core negative assessment stands. |
| oxFa | 6 | 5 | While the authors addressed some specific queries, they failed to conduct the critical head-to-head comparison with closely related works like Speculative Rejection. This omission directly impacts the perceived novelty and contribution of the paper, likely leading to a lower score. |
| kr89 | 4 | 5 | The authors partially addressed the concern about related work by adding new experiments combining ES-CoT with other prompt methods. This positive addition may slightly improve the reviewer's view of the paper's completeness, though the integration of the theoretical framework may still be seen as somewhat isolated. |

---

### Decision · Program_Chairs · 2026-01-26

Reject